# Bag of Baselines for Multi-objective Joint Neural Architecture Search and Hyperparameter Optimization

**Sergio Izquierdo**                                    IZQUIERD@CS.UNI-FREIBURG.DE
**Julia Guerrero-Viu**                                  GUERRERO@CS.UNI-FREIBURG.DE
**Sven Hauns**                                          HAUNSS@TF.UNI-FREIBURG.DE
**Guilherme Miotto**                                    ALESSANG@CS.UNI-FREIBURG.DE
**Simon Schrodi**                                       SCHRODI@CS.UNI-FREIBURG.DE
**André Biedenkapp**                                    BIEDENKA@CS.UNI-FREIBURG.DE
*University of Freiburg*
**Thomas Elsken**                                       THOMAS.ELSKEN@DE.BOSCH.COM
*Bosch Center for Artificial Intelligence*
**Difan Deng**                                          DENG@TNT.UNI-HANNOVER.DE
**Marius Lindauer**                                     LINDAUER@TNT.UNI-HANNOVER.DE
*Leibniz University Hannover*
**Frank Hutter**                                        FH@CS.UNI-FREIBURG.DE
*University of Freiburg and Bosch Center for Artificial Intelligence*

## Abstract

While both neural architecture search (NAS) and hyperparameter optimization (HPO) have been studied extensively in recent years, NAS methods typically assume fixed hyperparameters and vice versa. Furthermore, NAS has recently often been framed as a multi-objective optimization problem, in order to take, e.g., resource requirements into account. In this paper, we propose a set of methods that extend current approaches to jointly optimize neural architectures and hyperparameters with respect to multiple objectives. We hope that these methods will serve as simple baselines for future research on multi-objective joint NAS + HPO.

## 1. Introduction

Neural architecture search (NAS) and hyperparameter optimization (HPO) are both important components of AutoML, but there exists little work on *joint* NAS and HPO. NAS methods typically assume fixed hyperparameter configurations and HPO fixed architectures, even though it seems natural that different architectures require different hyperparameter configurations to yield optimal performance. Indeed, there is evidence that this is the case. For example, Gastaldi (2017) showed that the strongest version of the proposed shake-shake regularization performs best for some architectures, but is too strong for other architectures, resulting in poor performance or even divergence during training. Thus, a joint optimization of hyperparameter configurations and architecture can be expected to be beneficial.

Furthermore, while NAS and HPO methods typically optimize for accuracy, in many real-world applications there is more than one objective. Common objectives next to accuracy are, e.g., memory requirements, energy consumption or latency on the target hardware where the neural network is eventually deployed. In this paper, we take a first step in the

direction of *multi-objective joint NAS + HPO* by proposing and empirically evaluating a set of simple, yet powerful baseline methods. All our baseline methods essentially extend current NAS or HPO approaches to cover both classical and architectural hyperparameters, optimized under multiple objectives; see Table 1 for an overview of the methods we propose.

## 2. Related Work and Background

**Neural Architecture Search (NAS).** NAS refers to the task of learning neural network architectures from data (Elsken et al., 2019b; Wistuba et al., 2019). NAS approaches often employ black-box optimization methods, such as evolutionary algorithms (Real et al., 2017, 2019), reinforcement learning (Zoph and Le, 2017), or Bayesian optimization (Mendoza et al., 2016; Kandasamy et al., 2018). However, due to the large computational costs, researchers have developed methods tailored towards NAS, e.g., (gradient-based) optimization on one-shot models (Bender et al., 2018; Pham et al., 2018; Liu et al., 2019).

**Hyperparameter Optimization (HPO).** The field of HPO (see, e.g., (Feurer and Hutter, 2019)) automates the search for well performing hyperparameter configurations. Bayesian Optimization (BO) (Brochu et al., 2010; Shahriari et al., 2016) is a popular framework for solving HPO problems by employing cheap-to-evaluate surrogate model for predicting the performance of hyperparameter configurations and using an acquisition function to trade-off exploration and exploitation for selecting a new candidates. Evolutionary methods are also a common choice for HPO (Loshchilov and Hutter, 2016; Jaderberg et al., 2017). For this, a population of hyperparameter configurations is evolved over time by mutating or crossing well-performing candidates.

*Multi-fidelity methods* are also often employed as a tool for HPO to speed up function evaluations. Successive Halving (SH) (Jamieson and Talwalkar, 2016) and Hyperband (HB) (Li et al., 2018) are two powerful multi-fidelity strategies that allocate more budget on the well-performing hyperparameter configurations and achieve strong anytime performance. However, both strategies select new configurations at random without exploiting the knowledge gained about well-performing regions. BOHB (Falkner et al., 2018), which combines BO and HB, overcomes this issue by guiding Hyperband via a TPE model.

**Joint NAS + HPO.** Few researchers so far have considered the joint optimization of architectures and hyperparameter configurations. Domhan et al. (2015) and Mendoza et al. (2016) use SMAC (Hutter et al., 2011) to jointly optimize both architectures and hyperparameter configurations, Hundt et al. (2019) use BO with Gaussian processes. Zela et al. (2018), Runge et al. (2019) and Zimmer et al. (2021) employed BOHB (Falkner et al., 2018) to achieve the same goal, and Awad et al. (2021) proposed DEHB for this problem. Saikia et al. (2019) optimized architecture and hyperparameters in two stages: they first employ DARTS (Liu et al., 2019) to search for better architectures for disparity estimation and then optimize the hyperparameters of the resulting architecture with BOHB. Finally, Dong et al. (2020) extended NAS methods using one-shot models to also consider hyperparameters.

**Multi-objective Optimization.** Multi-objective optimization (e.g., Miettinen (1999)) deals with the problem of minimizing multiple objective functions $f_1(\lambda), \dots, f_n(\lambda)$. In general, there is no single configuration $\lambda$ that minimizes all objectives since the objectives are typically contradicting. Rather, there are multiple *Pareto-optimal* solutions, meaning

that one cannot reduce any $f_i$ without increasing at least one other $f_j$ $(i \neq j)$. The set of Pareto-optimal solutions is called the *Pareto front.*

One class of algorithms for solving multi-objective problems are evolutionary algorithms. Criteria for selecting candidates being mutated and defining the best current solutions are typically based on *non-dominated sorting* (NDS) (Srinivas and Deb, 1994; Deb et al., 2002) and the *hypervolume indicator* (Emmerich et al., 2005; Beume et al., 2007; Bader and Zitzler, 2011). *NDS* extends the ranking of a set of candidates based on a single objective to multiple objectives, please refer to Appendx A.1 for more details. The *hypervolume indicator* of a population measures, informally speaking, the space of objective function values covered by the population; thus maximizing the hypervolume indicator corresponds to improving the Pareto front. Based on the hypervolume indicator, the *hypervolume subset selection problem* (HSSP) (Bader and Zitzler, 2011) is defined as the problem of finding a subset of the population so that the hypervolume is maximized for this subset. The HSSP can also be employed to identify candidates that contribute little to the hypervolume and thus can be considered poor, see Appendx A.2 for a more details.

BO approaches have been extended to multi-objective problems. For example, *expected hypervolume improvement* (EHVI) (Emmerich, 2005) extends the work by Mockus et al. (1978) by considering improvements to the Pareto front. Ozaki et al. (2020) extended TPE (Bergstra et al., 2011) to multi-objective TPE (MOTPE). In concurrent work, Salinas et al. (2021) and Schmucker et al. (2021) extend Hyperband and Asynchronous Successive Halving (Li et al., 2020), respectively, with NDS and a multi-objective candidate selection scheme. Both approaches ares closely related to `MO-BOHB` (Section 3.2). Multi-objective optimization also naturally arise in NAS since many real-world applications require efficient architectures w.r.t., e.g., energy consumption or latency. Consequently, a line of research frames NAS either as a constrained (Tan et al., 2019; Cai et al., 2019) or multi-objective (Elsken et al., 2019a; Lu et al., 2020) optimization problem.

## 3. Proposed Methods

In the following, we propose five simple, yet powerful extensions of existing HPO and NAS optimization techniques to multi-objective joint HPO + NAS.

### 3.1 `SH-EMOA`: Speeding up Evolutionary Multi-Objective Algorithms (EMOA)

The flexibility and conceptual simplicity of evolutionary algorithms make them directly applicable to multi-objective optimization problems. For example SMS-EMOA (Beume et al., 2007) evaluates the performance of each candidate based on its contribution to the dominated hypervolume. Although effective, evolutionary algorithms tend to be very sample-inefficient, making them computationally quite expensive. In order to deal with this problem, we propose `SH-EMOA` to speed up EMOAs by using a multi-fidelity approach based on successive halving (please see Algorithm 1 in the appendix for full details).

In a nutshell, we iterate EMOA by doubling the training budgets in each iteration, while the number of candidates is halved. Thus, many candidates are evaluated with a small budget to cover a wide range of solutions, while only well-performing candidates proceed to the next stage, and are evaluated with the next higher budget and used to generate new candidates. We use NDS and the HSSP to identify poorly performing candidates.

### 3.2 `MO-BOHB`: Generalization of BOHB to an Arbitrary Number of Objectives

In order to extend BOHB to multi-objective optimization, we make two modifications. Firstly, we replace TPE (Bergstra et al., 2011) originally used in BOHB by MOTPE (Ozaki et al., 2020) for selecting new configurations w.r.t. multiple objectives. Secondly, we extend HB in a similar fashion as for `SH-EMOA` and MOTPE to decide with which configuration to proceed in the next stage: we use NDS and the result of the HSSP; see Algorithms 2 and 3.

### 3.3 `MS-EHVI`: Mixed Surrogate Expected Hypervolume Improvement

Although EHVI can be directly applied for joint NAS and HPO obtaining competent results, we further enhance the algorithm by a simple observation from Elsken et al. (2019a): while some of the objective functions are expensive to evaluate (e.g., evaluating the accuracy is expensive since it requires training the network first), others are cheap to evaluate (e.g., the number of parameters). Thus, rather than relying on a surrogate model for *every* objective function as in vanilla EHVI, we solely use surrogate models for the expensive objective function and directly evaluate the cheap objectives. This way, we avoid fitting surrogate models for objectives which are cheap to evaluate anyway. We refer to Algorithm 4.

### 3.4 `MO-BANANAS`

BANANAS (White et al., 2021) uses neural networks as a performance predictor of candidates within BO. To extend BANANAS for multi-objective optimization, we employ crowd sorting (Raquel and Jr, 2005) in combination with independent Thompson sampling to select parents used to generate new candidates by mutating the parents. Based on the performance predictor, the next candidates are chosen - again based on independent Thompson sampling, see Algorithm 5. We furthermore employ successive halving to quickly discard poorly-performing architectures.

### 3.5 `BULK & CUT`

`BULK & CUT` combines a simple evolutionary strategy with BO. The name `BULK & CUT` comes from the fact that the algorithm first looks for high accuracy models by successively enlarging them with network morphisms (Chen et al., 2016), then shrinking them using pruning techniques in combination with knowledge distillation (Hinton et al., 2015). `BULK & CUT` comprises three sequential phases:(i) initialization: sample random architectures, (ii) bulk-up: generate offsprings by applying network morphisms and (iii) cut-down: prune models.

After the initialization phase is completed, parents are selected for the bulk-up phase. For this, we propose the *Paretsilon Greedy* criterion, which combines non-dominated sorting and an $\epsilon$-greedy exploration strategy, as described in Algorithm 6. Once a parent is chosen, an offspring is generated by applying a network morphism (a commonly used operation in the NAS literature that avoids costly retraining from scratch (Elsken et al., 2019a; Cai et al., 2018)). In the cut-down phase, we use structured pruning (Anwar et al., 2017) to shrink models, i.e., eliminating units from fully-connected layers and filters from convolutional layers, instead of dropping individual weights. We employ knowledge distillation (Hinton et al., 2015) for training the shrunken models so that they match their parent's output (Elsken et al., 2019a; Prakosa et al., 2020; Chen et al., 2021). During all phases,

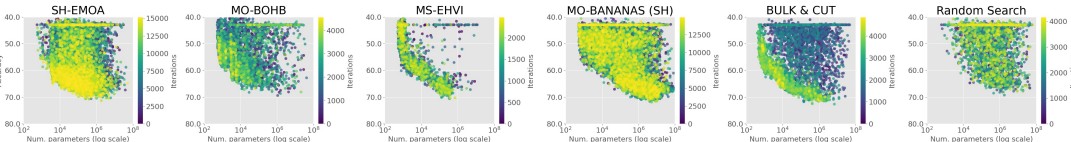

Figure 1: Sampled configurations for each method on Flowers dataset.

the non-architectural hyperparameters (e.g., learning rate and weight decay) are optimized via constrained BO. The term *constrained* here refers to the fact that the optimization of the acquisition function is performed with constrains on the already chosen architectural hyperparameters. Algorithm 7 in the appendix summarizes how `BULK & CUT` works.

## 4. Experiments

We used the Oxford-Flowers dataset (Nilsback and Zisserman, 2006) with images scaled down to a resolution of 16x16 for computational reasons. Each proposed method, along with random search as the simplest baseline, was run 10 times for 24 hours each. A maximum budget of 25 epochs for training is defined, although it is up to the methods to decide if they want to train with a smaller budget to speed up the search. We search over a variable number of convolutional layers (followed by ReLU and max-pooling), with a variable number of filters and kernel size each as well as optional batch normalization. A global average pooling may be applied afterwards, followed by a variable number of fully connected layers. We use Adam (Kingma and Ba, 2015) to optimize neural network weights, with a searchable learning rate and batch size. We refer to Table 2 for a summary of our search space. We furthermore evaluated the proposed methods on Cifar-10 on the NAS-Bench-201 (Dong and Yang, 2020) search space (except `BULK & CUT` since it is not directly applicable to the benchmark). NAS-Bench-201 does not contain any optimizable hyperparameters, however comes with a more interesting NAS search space and furthermore allows inspecting the true Pareto front due to the tabular nature of the benchmark.

We target network size (by means of number of parameters) and classification accuracy as the objectives of our multi-objective optimization. We split the datasets into training, validation and test splits. The splits are used for training model parameters, NAS + HPO and final evaluation, respectively.

### 4.1 Results

**Visualizing sampled configurations.** Figure 1 visualizes the sampled configurations for each method across all 10 random seeds on Flowers. All methods (except random search) explore the Pareto front, which is what they were designed for. However, they also significantly differ in the exploration strategy. `SH-EMOA` explores both objective functions equally well but still samples poor configurations in later iterations. `MO-BOHB` tends to focus on smaller networks in later iterations, while `MS-EHVI` very quickly discovers networks close to the Pareto front and mostly samples new configures there. After an initial phase, `BULK & CUT` also mostly samples candidates close to the Pareto front in later iterations.

**Performance comparison.** Figure 2 (a,b,d,e) shows the hypervolumes over time on Flowers and NAS-Bench-201, respectively. For Flowers, all our methods clearly outperform

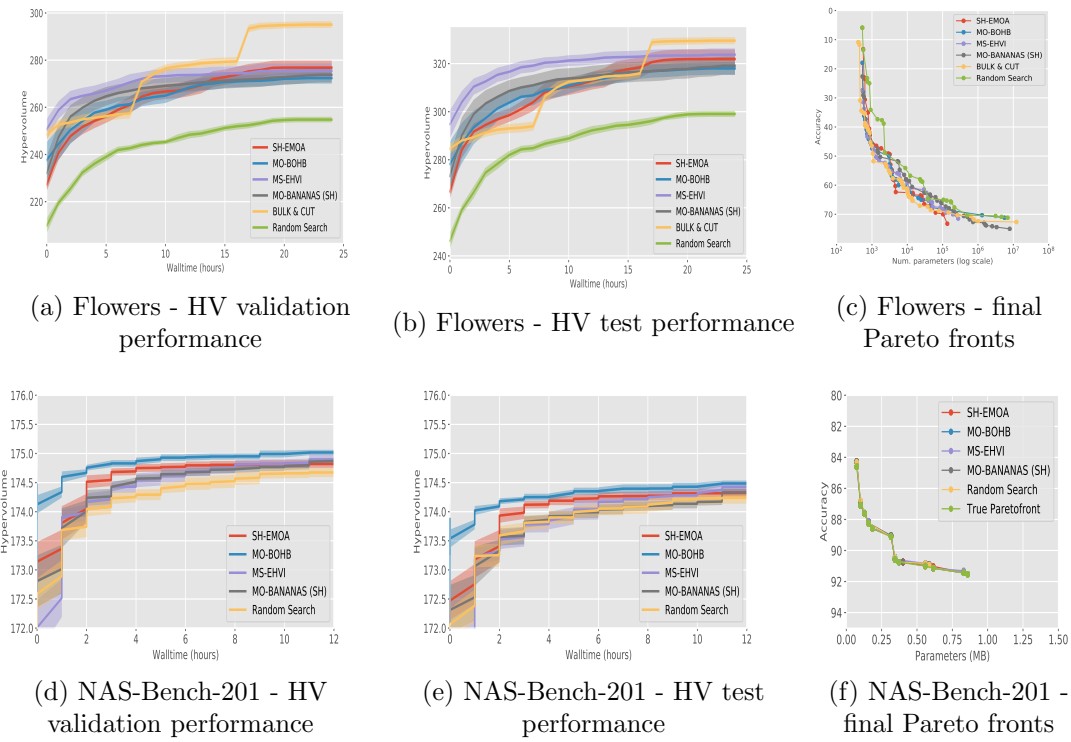

(a) Flowers - HV validation performance

(b) Flowers - HV test performance

(c) Flowers - final Pareto fronts

(d) NAS-Bench-201 - HV validation performance

(e) NAS-Bench-201 - HV test performance

(f) NAS-Bench-201 - final Pareto fronts

Figure 2: Hypervolume over time obtained by methods and final Pareto fronts. We show means ± SEM for hypervolume. First rows shows Flowers, second row NAS-Bench-201.

random search. `MS-EHVI` converges very fast but is eventually outperformed by `BULK & CUT`, which however performs less strongly in the initial phase. `SH-EMOA`, `MO-BOHB` and `MO-BANANAS` perform similarly. On NAS-Bench-201, `SH-EMOA` and `MO-BOHB` outperform the other methods initially, but all methods eventually achieve a similar performance. Figure 2 (c, f) also shows the final Pareto fronts when combining the results from all seeds. For Flowers, the proposed methods perform similarly for the range of parameters from $10^3$ to $10^5$, but some methods have problems with covering smaller or larger models. When looking at results for each seed (Figure 3 in the appendix), we however also noticed that the results vary across seeds, indicating that initializing might have a high impact on the performance and that the budget of 24 hours might not be sufficient for the methods to converge or that they simply get stuck in a local optimum. On NAS-Bench-201, each method discovers a Pareto front very close to the true Pareto front, suggesting that the chosen budget is sufficiently large to explore the whole space.

## 5. Conclusions

We addressed the problem of joint hyperparameter optimization and neural architecture search under multiple objectives by extending existing methods to this scenario. We propose several methods serving as baselines for future research in this direction. To facilitate this, all our code is available at `https://github.com/automl/multi-obj-baselines`.

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

## Appendix A. Related work - extended

| Proposed Method | Based on | Extended by |
|---|---|---|
| `SH-EMOA` (Sec. 3.1) | multi. obj. evolution | successive halving |
| `MO-BOHB` (Sec. 3.2) | BOHB | multi-objective for candidate selection, MOTPE |
| `MS-EHVI` (Sec. 3.3) | BO with EHVI | no surrogate for cheap objectives |
| `MO-BANANAS` (Sec. 3.4) | BANANAS | multi-objective candidate selection, successive halving |
| `BULK & CUT` (Sec. 3.5) | EA, BO | network morphism, pruning with knowledge distillation, constrained BO |

Table 1: Overview of the proposed methods.

### A.1 Non-dominated sorting

*NDS* extends the ranking of a set of candidates based on a single objective to multiple objectives $f = (f_1, \ldots, f_n)$ in the following way:

- Compute the Pareto front $\mathcal{F}_1 = pareto\_front(\mathcal{P}|f)$ of the current population $\mathcal{P}$ and assign all members of this Pareto front $\mathcal{F}_1$ the best rank.

- Remove the previous Pareto front from the population $\mathcal{P}$ and compute the Pareto front for the remaining population: $\mathcal{F}_2 = pareto\_front(\mathcal{P}\backslash\mathcal{F}_1|f)$. Members of this new Pareto front $\mathcal{F}_2$ are assigned the second best rank.

- Iterate this process until all members of the population have been assigned a rank.

Thus, a run of NDS partitions the population into sets $\mathcal{F}_1, \ldots, \mathcal{F}_k$, where a candidate $\lambda \in \mathcal{F}_i$ outperforms another candidate $\lambda' \in \mathcal{F}_j$ with respect to all objectives if $i < j$.

### A.2 Hypervolumne subset selection problem

The *hypervolume indicator* $I_H$ of a population measures, informally speaking, the space of objective function values covered by the population; maximizing the hypervolume indicator corresponds to improving the Pareto front and finding better solutions. Based on the hypervolume indicator, the *hypervolume subset selection problem* (HSSP) (Bader and Zitzler, 2011) is defined as the problem of finding a subset $\mathcal{P}_{HSPP} \subset \mathcal{P}$ of a certain size $k$ so that the hypervolume is maximized for this subset: $\mathcal{P}_{HSPP} \in \arg\max_{\mathcal{P}' \subset \mathcal{P}, |\mathcal{P}'|=k} I_H(\mathcal{P}')$. The HSSP can also be solved to identify a poorly performing candidate by setting $k = |\mathcal{P}| - 1$ and choosing the poor candidate $\lambda_{poor}$ as the one that gets removed from the population via HSSP: $\{\lambda_{poor}\} = \mathcal{P} \setminus \mathcal{P}_{HSPP}$. We refer to Bader and Zitzler (2011) for a more formal introduction.

## Appendix B. Full Details on the Various Baselines

### B.1 Implementation details on `SH-EMOA`

- Population size and total number of samples: In Flowers dataset, we use $s_{\mathrm{pop}} = 100$ and $n_{\mathrm{fe}}^{total} = 15000$.

---

**Algorithm 1:** `SH-EMOA`

---

**Input** : number of function evaluations $n_{\text{fe}}^{total}$ , population size $s_{\text{pop}}$, maximum budget $b_{max}$, number of SH iterations $n$, objectives $f$

**Output:** Pareto front w.r.t. $f$

1 Generate initial population $\mathcal{P}$ of size $s_{\text{pop}}$

2 $b \leftarrow \lfloor b_{max}/2^{n-1} \rfloor$                                      /* initial budget */

3 $n_{\text{fe}} \leftarrow \lfloor n_{\text{fe}}^{total} / \sum_{i=0}^{n-1} 2^{-i} \rfloor$         /* number of FE for the initial budget */

4 **for** $i = 1$ **to** $n$ **do**

5     Evaluate $f(\lambda)$ for all $\lambda \in \mathcal{P}$ with budget $b$

6     **for** $j = 1$ **to** $n_{fe} - s_{pop}$    /* generate candidates for remaining FEs */

7     **do**

8        Generate new candidate $\lambda_{\text{new}}$    /* parent selection and variation */

9        Evaluate $f(\lambda_{\text{new}})$ on budget $b$

10        $[\mathcal{F}_1, ..., \mathcal{F}_k] \leftarrow \text{NDS}(\mathcal{P} \cup \{(\lambda_{\text{new}}, f(\lambda_{\text{new}}))\})$

11        $\lambda_{\text{poor}} \leftarrow \text{HSSP}(\mathcal{F}_k, |\mathcal{F}_k| - 1)$

12        $\mathcal{P} \leftarrow (\mathcal{P} \cup \{(\lambda_{\text{new}}, f(\lambda_{\text{new}}))\}) \setminus \{(\lambda_{\text{poor}}, f(\lambda_{\text{poor}}))\}$

13     $n_{\text{fe}} \leftarrow n_{\text{fe}}/2$                /* half number of FE in next budget */

14     $b \leftarrow 2b$                         /* double budget */

15 **return** $pareto\_front(\mathcal{P}|f)$

---

- Parent selection: We use tournament selection by randomly sampling $k$ potential parents from the current population (uniform distribution) and choose the parent with highest fitness. We use $k = 3$.

- Variation: On each step we choose either mutation or recombination strategy with equal probability. Mutation is defined as a uniformly distributed random variation of 5 hyperparameters from the parent configuration. For recombination, we use two parents and choose each hyperparameter from one of them with equal probability. As we have a conditional search space, relationships between hyperparameters are taken into account when a new individual is created. For example, if a mutation increases the total number of convolutional layers, the size of the kernel for each new layer is also added.

## B.2 Implementation details on `MO-BOHB`

Algorithms 2 and 3 show pseudo code for `MO-BOHB` and its sampling step, respectively. Note the close resemblance to the original BOHB (differences marked in red), as `MO-BOHB` generalizes BOHB to any number of objectives. Note however, that there are two minor differences between the current version of our proposed `MO-BOHB` and the original BOHB implementation: (i) `MO-BOHB` uses an hierarchy of one-dimensional KDEs, whereas BOHB use a single multi-dimensional KDE, and (ii) we do not multiply bandwidths by a constant factor $b_w$. In future versions of `MO-BOHB`, we suspect that using a single multi-dimensional KDE and multiplication of bandwidths may further improve performance by better handling interaction effects between (architectural) hyperparameters and encouraging more exploration

around promising configurations, respectively.

---

**Algorithm 2: MO-BOHB**

**Input** : budgets $b_{min}$ and $b_{max}$, configurations discarding factor $\eta \in \mathbb{N}_{>0}$, and objectives $f$

**Output:** Pareto front w.r.t. f

1 $s_{max} \leftarrow \lfloor \log_\eta \frac{b_{max}}{b_{min}} \rfloor$
2 $\mathcal{P}_b \leftarrow [\ ] \forall b \in \{\eta^{-s} \cdot b_{max} | s = s_{max}, s_{max-1}, ..., 0\}$
3 **while** *not stopping criterion* **do**
4    **for** $s \in \{s_{max}, s_{max-1}, ..., 0\}$ **do**
5       sample $n = \lceil \frac{s_{max}+1}{s+1} \eta^s \rceil$ configurations $\lambda_1, ..., \lambda_n$ using Algorithm 3
6       run modified SH on $\lambda_1, ..., \lambda_n$ with initial budget $\eta^{-s} \cdot b_{max}$
7       add observations $\{(\lambda_i, f(\lambda_i))\}$ of each budget $b$ to $\mathcal{P}_b$
8 **return** $pareto\_front(\mathcal{P}_{b_{max}} | f)$

---

**Algorithm 3: Sampling in MO-BOHB**

**Input** : observations $\mathcal{P}$, fraction of random runs $\rho$, quantile $\gamma$, number of samples $n$, and minimum number of points $N_{min}$ to build a model

**Output:** next configuration to evaluate

1 **if** $rand() < \rho$ **then**
2    **return** random configuration
3 $b \leftarrow \arg\max\{\mathcal{P}_b : |\mathcal{P}_b| \geq N_{min} + 2\}$
4 **if** $b = \emptyset$ **then**
5    **return** random configuration
6 greedily split $\mathcal{P}$ into good $\mathcal{P}_l$ or bad $\mathcal{P}_g$ observations using NDS & HSSP
7 fit KDEs $l$ and $g$ based on $\mathcal{P}_l$ or $\mathcal{P}_g$, respectively
8 draw $n$ samples according to $l(\lambda)$
9 **return** sample with highest ratio $\frac{l(\lambda)}{g(\lambda)}$

---

In all our experiments, we set the meta-parameters of MO-BOHB as follows: For the HB part of MO-BOHB, we set the configuration discarding factor of to $\eta = 3$, use minimum budget $b_{min} = 5$ and maximum budget $b_{max} = 25$. In the BO part of MO-BOHB, we use an random fraction $\rho = 1/6$, set the quantile to $\gamma = 0.1$, sample $n = 24$ configurations, and use a minimum of $N_{min} = 2 \cdot |HPs| + 1$ points before building a model.

## B.3 Details on MS-EHVI

---

**Algorithm 4: MS-EHVI**

**Input** : expensive and cheap objectives $f = (f_{\text{exp}}, f_{\text{cheap}})$, surrogate model $\hat{f}_{\text{exp}}$, number of function evaluations $n_{\text{fe}}$

**Output:** Pareto front w.r.t. $f$

1 Initialize population $\mathcal{P}$ with initial observations
2 **for** $k = 1$ **to** $n_{fe}$ **do**
3    Fit surrogate model $\hat{f}_{\text{exp}}$ on $\mathcal{P}$
4    Select next candidate: $\lambda_{\text{new}} \in argmax_\lambda \text{EHVI}(\lambda | \mathcal{P}, \hat{f}_{\text{exp}}, f_{\text{cheap}})$
5    Evaluate $f_{\text{exp}}(\lambda_{\text{new}})$
6    Update data: $\mathcal{P} \leftarrow \mathcal{P} \cup \left\{ (\lambda_{\text{new}}, f(\lambda_{\text{new}})) \right\}$
7 **return** $pareto\_front(\mathcal{P} | f)$

---

## B.4 Details on `MO-BANANAS`

We do not use the path-based encoding from White et al. (2021) since it is not meaningful for our search space. Rather, we employ a simple real-valued vector representation, which furthermore also directly allows us to include non-architectural hyperparameters. We employ Gaussian noise for mutating parents: We assume integer-valued hyperparameters (e.g., number of layers), and normalize each value by dividing by the maximum value to map each hyperparameter to the range $[0, 1]$: we then add Gaussian noise. Before each function evaluation, this continuous representation is discretized by choosing the integer-valued hyperparameter which is closed to the mutation value after normalization.

---

**Algorithm 5:** `MO-BANANAS`

**Input** : neural predictor $\hat{f}$, number of candidates to mutate $n_{\text{mut}}$, mutation variance $\sigma^2$, number of new candidates $n_{\text{new}}$, objectives $f$
**Output:** Pareto front w.r.t. $f$

1 Generate initial population $\mathcal{P}$
2 **for** $i = 1$ **to** $n$ **do**
3 $\quad$ train neural predictor $\hat{f}$ on $\mathcal{P}$
4 $\quad$ sort $\mathcal{P}$ using NDS($\mathcal{P}$) and crowdingDistance($\mathcal{P}$)
5 $\quad$ choose top-$n_{\text{mut}}$ candidates from $\mathcal{P}$ and **mutate** by adding noise $\eta \sim \mathcal{N}(0, \sigma^2)$
$\quad$ $\quad$ (drawn independently for each dimension of the candidates)
6 $\quad$ evaluate chosen $n_{\text{mut}}$ candidates using $\hat{f}$
7 $\quad$ choose top-$n_{\text{new}}$ candidates $\lambda_1, \ldots, \lambda_{n_{\text{new}}}$ via independent Thompson sampling
8 $\quad$ evaluate $f(\lambda_1), \ldots, f(\lambda_{n_{\text{new}}})$
9 $\quad$ $\mathcal{P} \leftarrow \mathcal{P} \cup \{(\lambda_1, f(\lambda_1)), \cdots, (\lambda_{n_{new}}, f(\lambda_{n_{new}}))\}$
10 **return** $pareto\_front(\mathcal{P}|f)$

---

## B.5 Details on `BULK & CUT`

---

**Algorithm 6:** Paretsilon greedy

**Input** : Population $\mathcal{P}$, exploration probability $\epsilon$
**Output:** $\lambda \in \mathcal{P}$

1 **while** *True* **do**
2 $\quad$ $\mathcal{F} \leftarrow pareto\_front(\mathcal{P})$;
3 $\quad$ **if** $rand() \leq 1 - \epsilon$ *or* $\mathcal{F} == \mathcal{P}$ **then**
4 $\quad$ $\quad$ Sample $\lambda$ from $\mathcal{F}$;
5 $\quad$ $\quad$ **return** $\lambda$;
6 $\quad$ **else**
7 $\quad$ $\quad$ $\mathcal{P} \leftarrow \mathcal{P} \setminus \mathcal{F}$;

---

---

**Algorithm 7:** `BULK & CUT`

---

**Input** : time budgets $T_1 < T_2 < T_3$, exploration probability $\epsilon$, objectives $f$
**Output:** set of Pareto optimal solutions

**1** $\mathcal{P} \leftarrow \emptyset$ //population set
**2** $t \leftarrow elapsed\_time()$
**3** **while** $t < T_3$ **do**
**4**    **if** $t \in [0, T_1)$ **then**
**5**       $\lambda_\alpha \leftarrow random\_architecture()$
**6**    **if** $t \in [T_1, T_2)$ **then**
**7**       $\lambda_\alpha \leftarrow paretsilon\_greedy(\mathcal{P}, \epsilon)$    `/* parent selection (see Alg. 6) */`
**8**       $\lambda_\alpha \leftarrow network\_morphism(\lambda_\alpha)$
**9**    **if** $t \in [T_2, T_3]$ **then**
**10**      $\lambda_\alpha \leftarrow paretsilon\_greedy(\mathcal{P}, \epsilon)$    `/* parent selection (see Alg. 6) */`
**11**      $\lambda_\alpha \leftarrow prune\_and\_distill\_knowledge(\lambda_\alpha)$
**12**   $\lambda_\beta \leftarrow constrained\_BO(\lambda_\alpha)$        `/* other hyperparameters */`
**13**   evaluate $f(\lambda_\alpha, \lambda_\beta)$
**14**   Update $constrained\_BO$ with $(\lambda_\alpha, \lambda_\beta, f(\lambda_\alpha, \lambda_\beta))$
**15**   $\mathcal{P} \leftarrow \mathcal{P} \cup \{(\lambda_\alpha, \lambda_\beta, f(\lambda_\alpha, \lambda_\beta))\}$
**16**   $t \leftarrow elapsed\_time()$
**17** **return** $pareto\_front(\mathcal{P})$

---

## Appendix C. Experimental details.

We used the Oxford-Flowers dataset (Nilsback and Zisserman, 2006), a small dataset composed of 17 different classes with 80 examples each, to show the performance of the proposed approaches in environments where many, cheap function evaluations are available. All images are scaled down to $16x16$ for computational reasons. We split the datasets as follows: we randomly split the data into 60% for training, 20% for validation and 20% for testing. Neural network weights are always trained on training data, their performance on validation data is used to guide hyperparameter and architecture optimization, and the test set is only used for evaluation.

## Appendix D. Supplemental Results

Figures 3 and 4 show all the Pareto fronts found with different seeds.

| Hyperparameter | Range | Log scale |
|---|---|---|
| Num. convolutional layers | $\{1, 2, 3\}$ | No |
| Num. filters conv. layer $i$ | $[2^4, 2^{10}]$ | Yes |
| Kernel size | $\{3, 5, 7\}$ | No |
| Batch normalization | $\{true, false\}$ | No |
| Global average pooling | $\{true, false\}$ | No |
| Num. fully connected layers | $\{1, 2, 3\}$ | No |
| Num. neurons FC layer $i$ | $[2^1, 2^9]$ | Yes |
| Learning rate | $[10^{-5}, 10^0]$ | Yes |
| Batch size | $[2^0, 2^9]$ | Yes |

Table 2: Joint space of architectural and non-architectural hyperparameters being optimized.

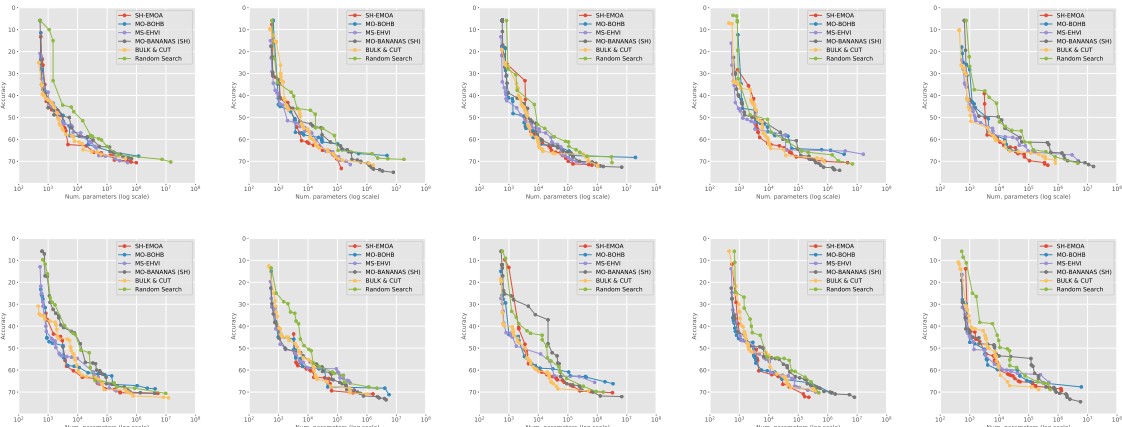

Figure 3: Pareto fronts obtained for different initial random seeds on Flowers dataset.

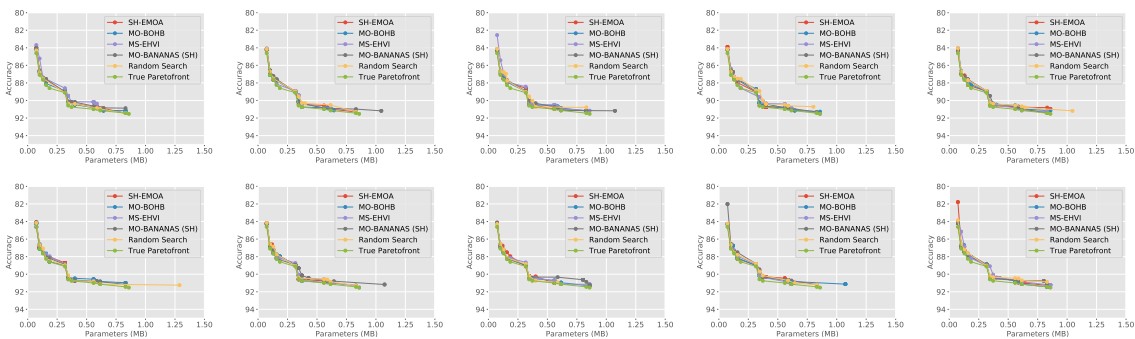

Figure 4: Pareto fronts obtained for different initial random seeds on NAS-Bench-201 dataset.

