# OpenReview forum: "Bag of Baselines for Multi-objective Joint Neural Architecture Search and Hyperparameter Optimization"
_ICML.cc/2021/Workshop/AutoML — AutoML@ICML2021 Poster_

### Official Review · Reviewer_HrEF · 2021-06-12
**A review of the paper: "Bag of Baselines for Multi-objective Joint Neural Architecture Search and Hyperparameter Optimization ".**

**Rating:** 7
**Confidence:** 4

**Review:**

General Comments

The paper introduces a new approach to combine approaches to search for optimal neural network architectures with techniques to optimize hyper-parameters. This area remains mostly unexplored. The main idea is well explained and technically sound; the paper is well written and well organized.
Overall, the proposed solutions are attractive, based on clever modifications to existing baseline techniques; the solutions add efficiency and transform some processes into multi-objective methods. There is a clear contribution along with the proposed improvements over current approaches.

Below are two points on the paper:

•	Many of the proposed solutions try to allocate more resources to promising candidates while taking resources away from candidates exhibiting poor performance; this may seem intuitively correct, but it may fail in some circumstances. For example, when the optimization function is characterized by many peaks and valleys, moving away from a poor-performance region may fail to identify isolated hard-to-detect areas where an optimal solution lies. It would be ideal to see the risk of filtering out candidates as the landscape of the optimization changes from smooth to very rough.

•	The reported sensitivity of the results to the initialization step needs in-depth analysis. The effect of the initialization may veil any conclusions on the effectiveness of the proposed methods. How can we assess the impact of each proposed solution while quantifying the impact of different parameter initializations?



Minor Comments

Section 3.1., page 3, should read: “…we iterate EMOA by doubling the training budgets…”

---

### Official Review · Reviewer_bNHH · 2021-06-16
**Overall a strong paper that sets up nice a novel baselines for a growingly complex and important field of research (NAS + HPO)**

**Rating:** 8
**Confidence:** 4

**Review:**

Quality:
The fields of NAS and HPO always needs very clear baselines, especially as proposed methods get more and more complex. This paper contributes high value to the increasingly popular field of combining together both NAS and HPO and setups up 5 new and principled baselines. Often the literature underemphasizes the importance of having good baselines, especially in fields where the methods grow in complexity.

Clarity:
The paper is very clearly written and easy to follow.

Originality / Significance:
This work proposes the most extensive set of baselines I have seen combining both NAS + HPO and has some slightly new algorithmic variants to test them. Overall this is a new set of baselines that will be of great use to future researchers and practitioners. Many papers have already pointed out how many NAS methods are not actually that much better than random search, so the significance of having these baselines for the even more complex task of NAS + HPO is very needed.

Pros:
- Propose 5 simple baseline methods for combining NAS + HPO that are principled and based on other methods people use for NAS or HPO.
- Well written and easy to follow
- Has a great related work section that does a good job of contextualizing the current methods used in the NAS and HPO space and how they relate to their 5 new proposed variants

Cons:
- Missing some related works combining NAS + HPO like: EfficientNetV2: Smaller Models and Faster Training
- Minor: Table 1 "Based on" and "Extended by" contents are unclear to people who do not know the acronyms already. It would make it easier to read if this was self-contained (e.g. write out BOHB, BO, EHVI, etc...) or came after the terms were already introduced.
- Would be good to have a slightly more rigorous task for running the NAS + HPO other than Oxford flowers (such as CIFAR).
- It would also be good to rerun with Oxford Flowers setup that does not have so much variance across seeds to better test out the HPO + NAS methods

---

### Decision · Program_Chairs · 2021-06-21

Accept (Poster)